# PrivyNet: A Flexible Framework for Privacy-Preserving Deep Neural Network Training

## Abstract

Massive data exist among user local platforms that usually cannot support deep neural network (DNN) training due to computation and storage resource constraints. Cloud-based training schemes provide beneficial services, but suffer from potential privacy risks due to excessive user data collection. To enable cloud-based DNN training while protecting the data privacy simultaneously, we propose to leverage the intermediate representations of the data, which is achieved by splitting the DNNs and deploying them separately onto local platforms and the cloud. The local neural network (NN) is used to generate the feature representations. To avoid local training and protect data privacy, the local NN is derived from pre-trained NNs. The cloud NN is then trained based on the extracted intermediate representations for the target learning task. We validate the idea of DNN splitting by characterizing the dependency of privacy loss and classification accuracy on the local NN topology for a convolutional NN (CNN) based image classification task. Based on the characterization, we further propose PrivyNet to determine the local NN topology, which optimizes the accuracy of the target learning task under the constraints on privacy loss, local computation, and storage. The efficiency and effectiveness of PrivyNet are demonstrated with CIFAR-10 dataset.

## 1 Introduction

With the pervasiveness of sensors, cameras, and mobile devices, massive data are generated and stored on local platforms. While useful information can be extracted from the data, the training process can be too computationally intensive that local platforms are not able to support. Cloud-based services provide a viable alternative to enable deep model training but rely on excessive user data collection, which suffers from potential privacy risks and policy violations.

To enable the cloud-based training scheme while simultaneously protecting user data privacy, different data pre-processing schemes are proposed. Instead of releasing the original data, transformed representations are usually generated locally and then, uploaded for the target learning tasks. For the intermediate representations to be effective, there are two requirements, i.e. utility and privacy. The utility requirement urges that the target learning task can be accomplished accurately based on the released representations, while the privacy requirement forces the leakage of private information to be constrained within a satisfiable range. Furthermore, the transformation scheme should also be flexible enough for platforms with different computation and storage capabilities and for different types of data, which can be either high dimensional and continuous, like videos or images, or discrete.

**Related Works** Privacy and utility trade-off has been one of the main questions in the privacy research. Different measures of privacy and utility are proposed based on the rate-distortion theory (Sankar et al., 2010; Rebollo-Monedero et al., 2010; du Pin Calmon & Fawaz, 2012), statistical estimation (Smith, 2011), and learnability (Kasiviswanathan et al., 2011). To actively explore the trade-off between privacy and utility, in recent years, many different transformations have been proposed. Syntactic anonymization methods, including $k$-anonymity (Sweeney, 2002), $l$-diversity (Machanavajjhala et al., 2007) and $t$-closeness (Li et al., 2007), are proposed to anonymize quasi-identifiers and protect sensitive attributes in a static database. However, syntactic anonymization is hard to apply to high-dimensional continuous data because quasi-identifiers and sensitive attributes become hard to define.

Differential privacy is proposed to provide a more formal privacy guarantee and can be easily achieved by adding noise (Dwork & Nissim, 2004; Dwork et al., 2006; 2014). However, because differential privacy only prevents an adversary from gaining additional knowledge by inclusion/exclusion of an individual data (Dwork et al., 2014), the total information leakage from the released representations is not limited (Hamm, 2015). Meanwhile, to achieve differential privacy, existing works (Shokri & Shmatikov, 2015; Abadi et al., 2016) usually require local platforms to get involved in the backward propagation process, which makes it hard to deploy them on lightweight platforms.

Non-invertible linear and non-linear transformations are also proposed for data anonymization. Existing linear transformations rely on the covariance between data and labels (Enev et al., 2012) or the linear discriminant analysis (LDA) (Whitehill & Movellan, 2012) to filter the training data. However, linear transformations usually suffer from limited privacy protection since the original data can be reconstructed given the released representations. Recently proposed nonlinear transformations based on minimax filter (Hamm, 2015) or Siamese networks (Ossia et al., 2017) can provide better privacy protection. However, they can only be applied to protect privacy in the inference stage since iterative training scheme is required between the cloud and local platforms, for which privacy loss becomes very hard to control.

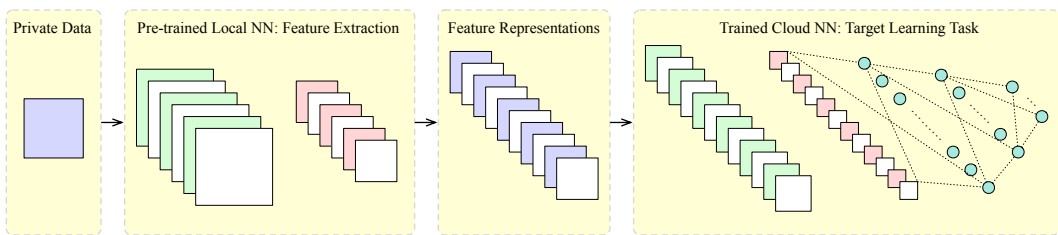

Figure 1: The proposed PrivyNet framework: the local NN is derived from pre-trained NNs for feature extraction and the cloud NN is trained for the target learning task. Privacy and utility trade-off is controlled by the topology of the local NN.

**Contribution** To this end, we propose PrivyNet, a flexible DNN training framework to achieve a fine-grained control of the trade-off between privacy and utility. PrivyNet divides a DNN model into two parts and deploys them onto the local platforms and the cloud separately. As shown in Figure 1, the local NN is used to generate intermediate representations while the cloud NN is trained for the learning task based on the released intermediate representations. The privacy protection is achieved through the transformation realized by the local NN, which is non-linear and consists of different lossy operations, including convolution, pooling, and so on. To avoid local training, we derive the local NN from pre-trained NNs. Our key insight here is that the initial layers of a DNN are usually used to extract general features that are not application specific and can enable different learning tasks. Therefore, by deriving the local NN from pre-trained NNs, a good utility can be achieved since useful features are embedded in the released representations, while privacy can be protected by selecting the topology of the local NN to control the specific features to release. Our main contributions are summarized as follows:

- We propose PrivyNet, a novel framework to split DNN model to enable cloud-based training with a fine-grained control of privacy loss.

- We characterize the privacy loss and utility of using CNN as the local NN in detail, based on which three key factors that determine the privacy and utility trade-off are identified and compared.

- A hierarchical strategy is proposed to determine the topology of the local NN to optimize the utility considering constraints on local computation, storage, and privacy loss.

- We verify PrivyNet by using the CNN-based image classification as an example and demonstrate its efficiency and effectiveness.

## 2 UTILITY AND PRIVACY CHARACTERIZATION FOR THE NN-BASED LOCAL TRANSFORMATION

In this section, we validate the idea of leveraging pre-trained NN for intermediate represetation generation by a detailed utility and privacy characterization. We use CNN-based image classification as an example. The overall characterization flow is illustrated in Figure 2. Given original data, feature represetations are first generated by the feature extraction network (FEN), which is selected from a pre-trained NN. Then, an image classification network (ICN) is trained based on the feature representations and the labels for the target learning task and an image reconstruction network (IRN) is trained to reconstruct the original images from the features. We measure the utility by the accuracy of the target learning task and the privacy by the distance between the reconstructed images and the original images. Here, it should be noted that when we train the IRN, we assume both the original images and the feature representations are known while the transformation, i.e. FEN, is unknown. This is aligned with our adversarial model, which will be described in detail in Section 3 and 4.

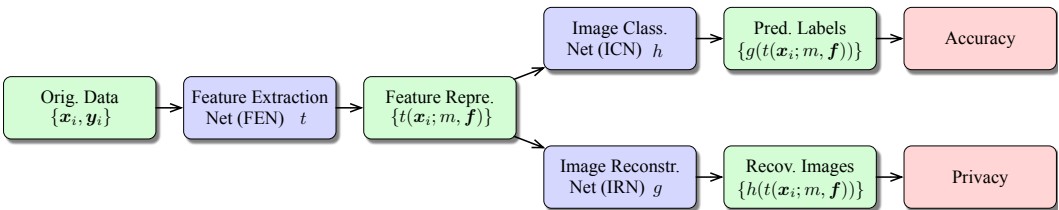

Figure 2: Privacy and utility characterization flow.

Formally, consider a collection of $N$ training instances $\mathcal{D} = \{(\boldsymbol{x}_i, \boldsymbol{y}_i)\}_{i=1}^N$. $\boldsymbol{x}_i \in \mathbb{R}^{W \times H \times D}$ represents the $i$-th image with $D$ channels and the dimension for each channel is $W \times H$. $\boldsymbol{y}_i \in \{0,1\}^K$ is the label indicator vector with $y_{i,k} = 1$ if $k$ is the correct label for the $i$-th image and $y_{i,k} = 0$ otherwise. Let $t : \mathbb{R}^{W \times H \times D} \rightarrow \mathbb{R}^{W' \times H' \times D'}$ be the transformation induced by the FEN. The depth of the output feature representations is $D'$ while the dimension for each feature is $W' \times H'$. $t$ is parameterized by the number of FEN layers $m$ and the subset of filters selected for each layer $\boldsymbol{f}$. Let $t(\boldsymbol{x}_i; m, \boldsymbol{f})$ denote the output representations for $\boldsymbol{x}_i$. For the $j$-th channel of the output representations for image $\boldsymbol{x}_i$, let $\boldsymbol{z}_i^j = t_j(\boldsymbol{x}_i; m, \boldsymbol{f})$ denote the corresponding flattened column vector.

**Utility** To evaluate the utility, given a collection of transformed representations $\{t(\boldsymbol{x}_i; m, \boldsymbol{f}), \boldsymbol{y}_i\}_{i=1}^N$, we learn a classifier $h^* : \mathbb{R}^{W' \times H' \times D'} \rightarrow \{0,1\}^K$ with minimized empirical risk for the target learning task, i.e.,

$$h^* = \operatorname{argmin}_h \sum_{i=1}^N \ell_u(h(t(\boldsymbol{x}_i; m, \boldsymbol{f})), \boldsymbol{y}_i), \qquad (1)$$

where loss function $\ell_u = 1$ if $h(t(\boldsymbol{x}_i; m, \boldsymbol{f})) \neq \boldsymbol{y}_i$ and $\ell_u = 0$ otherwise. We define the utility as the accuracy achieved by $h^*$,

$$Util := \frac{1}{N'} \sum_{i=1}^{N'} \ell_u(h^*(t(\boldsymbol{x}_i; m, \boldsymbol{f})), \boldsymbol{y}_i). \qquad (2)$$

where $N'$ is the number of testing instances. Better accuracy achieved by $h^*$ implies better utility for the transformed representations.

**Privacy** To evaluate the privacy, given $\{t(\boldsymbol{x}_i; m, \boldsymbol{f}), \boldsymbol{x}_i\}_{i=1}^N$, learn a reconstruction model $g^* : \mathbb{R}^{W' \times H' \times D'} \rightarrow \mathbb{R}^{W \times H \times D}$ that minimizes the distance between the reconstructed images and the original images, i.e.,

$$g^* = \operatorname{argmin}_g \frac{1}{N} \sum_{i=1}^N \ell_p(g(t(\boldsymbol{x}_i; m, \boldsymbol{f})), \boldsymbol{x}_i), \qquad (3)$$

where loss function $\ell_p$ is defined based on the pixelwise Euclidean distance

$$\ell_p(g(t(\boldsymbol{x}_i; m, \boldsymbol{f})), \boldsymbol{x}_i) = \| g(t(\boldsymbol{x}_i; m, \boldsymbol{f})) - \boldsymbol{x}_i \|_2^2 . \qquad (4)$$

We measure the privacy loss of the transformed representations by the peak signal-to-noise ratio (PSNR) of the reconstructed images compared to the original images

$$Priv := \frac{1}{N'} \sum_{i=1}^{N'} \text{PSNR}(g^*(t(\boldsymbol{x}_i; m, \boldsymbol{f})), \boldsymbol{x}_i). \quad (5)$$

Larger PSNR implies larger privacy loss from the transformed representations. To understand the implication of the PSNR values, we show the reconstructed images with different PSNRs in Appendix F.

We now characterize the impact of FEN topology on the privacy and utility of the transformed representations. The characterization settings are described in Appendix B in detail. As an example, we derive the FEN from VGG16 (Simonyan & Zisserman, 2014), which is pre-trained on Imagenet dataset (Russakovsky et al., 2015). We use CNN to construct $h$ for the image classification task and $g$ for the reconstruction task. The architectures of VGG16, ICN, and IRN are shown in Appendix B. The FEN topology is mainly determined by three factors, including the number of layers, the depth of the output channels, and the subset of channels selected as the output. In this section, we evaluate and compare each factor, which becomes the basis for the PrivyNet framework.

## 2.1 IMPACT OF THE NUMBER OF FEN LAYERS AND OUTPUT DEPTH

To evaluate the impact of the number of FEN layers and the output depth, we change the topology of the FEN to generate different sets of representations. Based on the generated representations, ICN and IRN are trained to evaluate the utility and privacy. We plot the change of utility and privacy as in Figure 3. As we can observe, while both utility and privacy get impacted by the change of the number of FEN layers and output depth, they show different behaviors. For privacy loss, we observe smaller PSNR of the reconstructed images, i.e., less privacy loss, either with the reduction of the output depth or the increase of FEN layers. For the change of accuracy, i.e., utility, when the number of FEN layer is small, it shows small degradation with the reduction of the output depth. Similarly, when the output depth is large, accuracy also remains roughly the same with the increase of the FEN layer. However, when the number of FEN layer is large while at the same time, the output depth is small, large accuracy degradation can be observed.

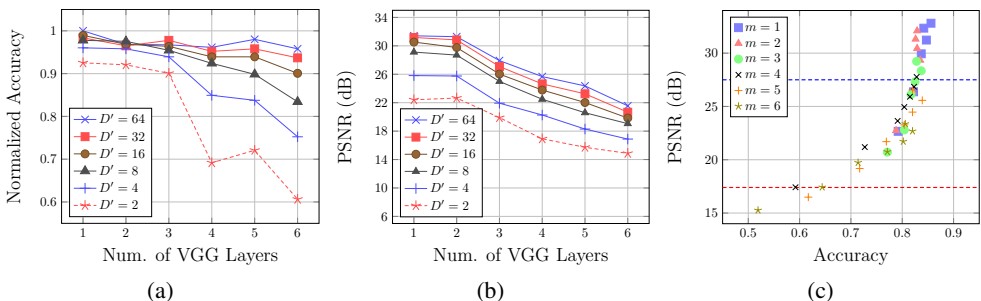

Figure 3: Impact of FEN topology on utility and privacy: (a) dependency of utility; (b) dependency of privacy; (c) utility and privacy trade-off.

We show the trade-off between the accuracy and PSNR in Figure 3(c). From Figure 3(c), we have two key observations, which become an important guidance for the PrivyNet framework:

- When the privacy loss is high (blue line, large PSNR), for the same privacy level, FEN with different topologies have similar utility.
- When the privacy loss is low (red line, small PSNR), for the same privacy level, FEN with more number of layers tends to provide better utility.

## 2.2 IMPACT OF OUTPUT CHANNEL SELECTION

Besides the number of FEN layers and the output channel depth, the selected subset of output channels will also impact the privacy and utility of the output representations. To understand the impact of

channel selection, we first compare the utility and privacy loss for the transformed representations induced by each single channel. As an example, we characterize the utility and privacy loss for the representations generated by each channel when the FEN consists of 4 VGG16 layers. The characterization result is shown in Figure 4(a). We also put the detailed statistics on utility and privacy in Table 4(c). As we can see, when $m = 4$, the utility achieved by the best channel is around 4X of that of the worst channel. Meanwhile, the privacy loss for the best channel is around 6 dB less compared to that of the worst channel. Large discrepancy can also be observed when we use 6 VGG16 layers to generated the FEN as in Figure 4(b).

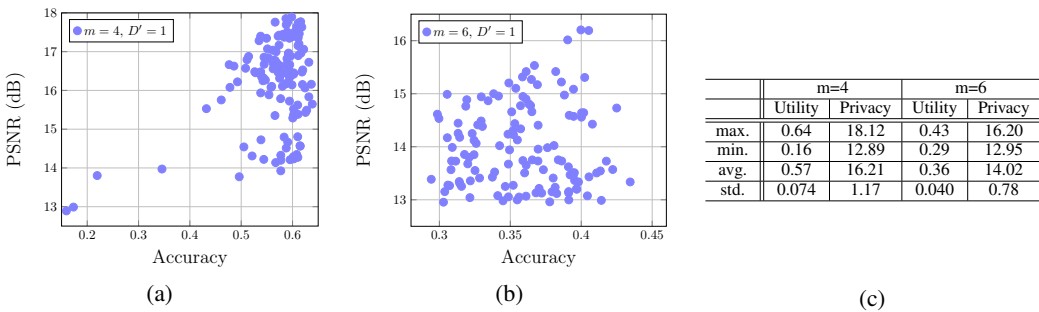

Figure 4: Difference on utility and privacy for single channel in the 4th and 6th VGG16 layer.

We then compare the impact of output channel selection with the impact of the number of FEN layers and output depth. We first fix the output channel depth to 2 and 8 and change the number of FEN layers. We then fix the layer of FEN to be 4 and 6 and change the output depth. For each setting, we randomly select 20 different sets of output channels and evaluate the privacy and utility for each set of released representations. We show the change of privacy and utility in Figure 5. From the comparison, we can observe that compared with the output channel selection, both utility and privacy show larger dependence on the number of FEN layers and output channel depth.

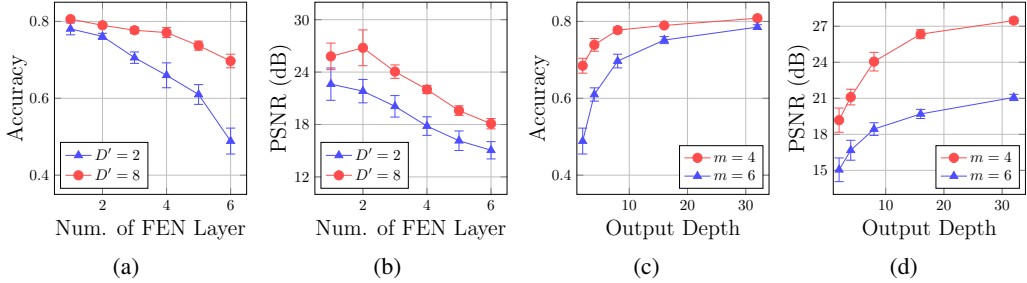

Figure 5: Comparison between the impact on utility and privacy of output channel selection and the number of FEN layers ((a) and (b)) and output depth ((c) and (d)).

## 2.3 MAJOR OBSERVATIONS

From the privacy and utility characterization of the representations generated from the pre-trained CNN-based transformation, we mainly have the following key observations:

- We can leverage the pre-trained CNN to build the FEN and explore the trade-off between utility and privacy by controlling the FEN topology.
- The privacy and accuracy trade-off can be controlled by the number of FEN layers, the output channel depth, and the output channel selection. Among the three factors, larger dependence can be observed for the first two factors for both privacy and utility.

Based on the observations above, in the next section, we propose our framework, PrivyNet, to determine the FEN topology in order to optimize the utility under the constraints on privacy as well as local computation capability and storage.

## 3   HIERARCHICAL STRATEGY TO DETERMINE FEN TOPOLOGY

In this section, we describe our framework, PrivyNet, to determine the topology of the FEN. In Section 2, the idea to derive the FEN from a pre-trained NN and control the trade-off between privacy and utility has been validated. In fact, besides the impact on privacy and utility, FEN topology also directly impacts the computation and storage on the local platforms. Especially for some lightweight platforms, like mobile devices, the constraints on the local computation and storage can be very important and must be considered when we design the FEN. To optimize the utility under the constraints on privacy, local computation, and storage, our PrivyNet framework is shown in Figure 6. In the first step, privacy characterization is carried out leveraging cloud-based services based on publicly available data. Performance profiling of different NNs is also conducted on local platforms. Then, based on the pre-characterization, the number of layers and the output depth of the FEN can be determined to consider the constraints on privacy, local computation capability, and storage. A supervised channel pruning step is then conducted based on the private data to prune the channels that are ineffective for the target learning task or cause too much privacy leakage. Then, the output channels are randomly selected, after which the FEN topology is determined.

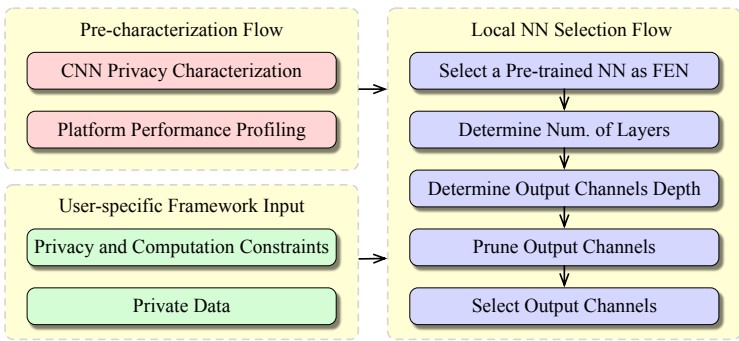

Figure 6: Overall flow of PrivyNet to determine the FEN topology.

### 3.1   ADVERSARIAL MODEL

Before introducing our framework, we first define the adversarial model. In this work, we assume the attackers have the released representations $\{t(\boldsymbol{x}_i; m, \boldsymbol{f})\}_{i=1}^N$ and the labels $\{\boldsymbol{y}_i\}_{i=1}^N$; and the original images $\{\boldsymbol{x}_i\}_{i=1}^N$. The assumption on the availability of the original images is much stronger than previous works (Hamm, 2015; Enev et al., 2012; Whitehill & Movellan, 2012). We believe such assumption is necessary because, on one hand, this enables us to provide a worst-case evaluation on the privacy loss induced by releasing the feature representations; while on the other hand, in many cases, we believe such accessibility to origin images is indeed possible. For example, the attackers may be able to inject some images into a database and get the corresponding representations generated by the FEN.

In our adversarial model, we also assume the transformation induced by the FEN to be unknown to the attackers. This is important as more sophisticated image reconstruction mechanisms can be available if the FEN is known by the attacker, which makes it very hard to evaluate and limit the privacy loss of the released representations. Because the FEN is derived from pre-trained NNs, both the architecture and weights of which may be available to the attackers as well, we need to protect the anonymity of the FEN. We have detailed description on FEN anonymity protection in Section 4.

### 3.2   PRE-CHARACTERIZATION FLOW

The pre-characterization stage mainly consists of two steps, including performance/storage profiling on local platforms and cloud-based privacy characterization for the pre-trained NNs. Performance and storage characterization is realized by profiling different pre-trained NNs on the local platforms. Such profiling is important because as described in Appendix D, different platforms have different computation capability and storage configurations, which directly determines the topology of the FEN applied on the platform.

The privacy characterization for the pre-trained NNs follows the process described in Section 2 and is realized by leveraging the cloud-based services. The reconstruction network is trained on publicly available data, which contains data of the same dimension and preferably from the same distribution. To verify the feasibility of such characterization, we do the same privacy characterization for different datasets, i.e., CIFAR-10 and CIFAR-100, and compare the PSNR for FEN with different topologies. As shown in Figure 7(b) and 7(c), very similar PSNR change can be observed for FEN with different topologies. We also run experiments to determine the number of samples required for the characterization. As shown in Figure 7(a), with data augmentation, less than $1000$ samples are needed to enable an accurate characterization. Moreover, because the privacy characterization is not the target learning task, it is acceptable for the detailed PSNR values to be less accurate, which helps to further reduce the requirement on the training samples.

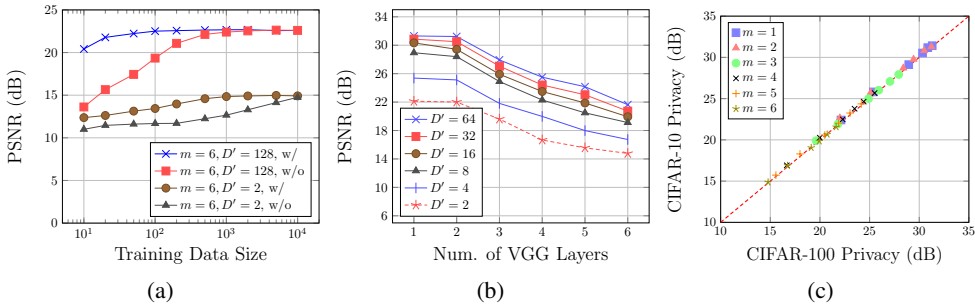

Figure 7: Privacy characterization: (a) small amount of samples are needed for the characterization with data augmentation; (b) and (c) very similar privacy characterization results can be acquired on different datasets, which indicates IRN can be trained on public data.

### 3.3 NUMBER OF LAYER AND OUTPUT CHANNEL DEPTH DETERMINATION

Now, we want to determine the topology for the FEN. Recall from Section 2 that compared with output channel selection, the number of FEN layers and the output channel depth have larger impacts on the privacy and accuracy of the generated representations. Therefore, in PrivyNet, we first determine these two factors. We rely on the results from the pre-characterization and determine the two factors considering the constraints on local computation, storage, and privacy loss. In Figure 8(a) and 8(b), we show the relation between the privacy and the local computation and storage on a mobile class CPU when we change the FEN topology. Based on Figure 8 and our observations in Section 2, we have the following strategy:

- When the privacy requirement is high, i.e., a small PSNR is required for the reconstructed images, because a FEN with more layers tends to provide a better utility, we select the deepest layer for the FEN following the constraints on computation and storage. Then, the output depth is determined by the privacy constraints.

- When the privacy requirement is low, i.e., a large PSNR is allowed, because FENs with different topologies tend to provide a similar utility, we just select a shallow FEN that can achieve the required privacy level and then determine the output channel depth according to the privacy constraints. This helps to achieve less local computation and storage consumption.

For example, assume the allowed PSNR is $28$ dB (large privacy loss is allowed, blue line in Figure 8(a)), while such privacy loss requirement can be achieved by FENs with different layers and output depths, we choose a shadow FEN, i.e., $m = 1$ and $D' = 4$, in order to minimize the local computation and storage. If the allowed PSNR is $17$ dB (low privacy loss is required, red line in Figure 8(a)), then, although FENs with 4, 5, and 6 layers can all achieve the privacy requirement, we choose $m = 6$ and $D' = 4$ in order to get a better utility.

## 3.4 OUTPUT CHANNEL SELECTION BY SUPERVISED PRUNING

After determining the number of layers and the output depth, we need to do the output channel selection. In Figure 4, we observe large discrepancies in utility and privacy for a single channel. Similar difference on privacy and utility can also be observed when we change the layers or increase the output channel depth as shown in Figure 9. Therefore, directly selecting output channels from the whole set suffer from large variance on utility and privacy and may result in the situation when a poor utility is achieved with large privacy leakage (upper left corner in Figure 9), which indicates the necessity of channel pruning.

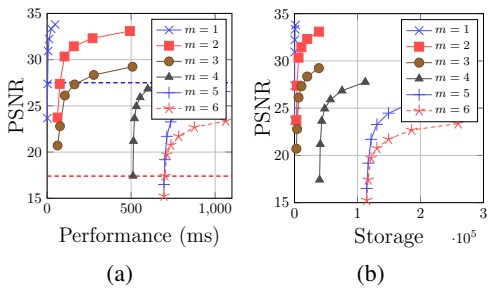 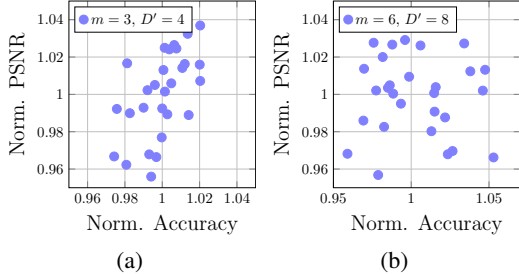

Figure 8: Relation between privacy and performance/storage with FEN topologies.

Figure 9: Difference on utility and privacy induced by random output channel selection.

Meanwhile, from Figure 4, we also observe that for a single channel, its utility and privacy loss are not correlated. We demonstrate the negligible correlation in Figure 10. According to Figure 10, when the layer of FEN is 6, among the top 32 channels that suffer from the largest privacy loss, 4 channels are among the 32 channels with the worst utility. The negligible correlation can also be observed for different output channel depths and FEN layers in Figure 9. The observation is important as it enables us to optimize the utility while suppressing the privacy loss simultaneously.

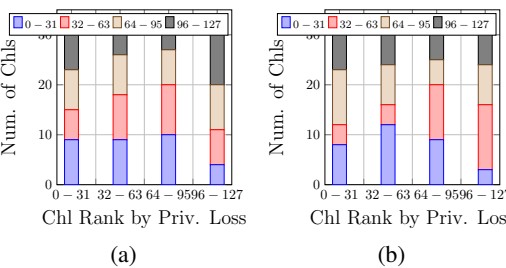 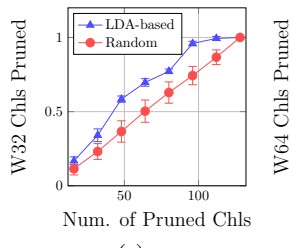 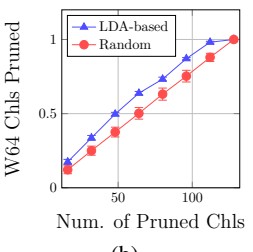

Figure 10: Utility and privacy relation for single channel: (a) $m = 4$, (b) $m = 6$.

Figure 11: Comparison with random strategy on pruning the worst (a) 32 and (b) 64 channels.

In our channel pruning process, we consider both utility and privacy. The privacy loss for each channel can be acquired from the offline pre-characterization, which can help to prune the channels with the largest privacy loss. To identify channels with the worst utility, we choose to leverage Fisher's linear discriminability analysis (LDA) (Fisher, 1936). The intuition behind Fisher's LDA scheme is that for an output channel to be effective, the distance of the representations generated by the channel for different images within the same class should be small. Meanwhile, the distance for representations among different classes should be large. In Fisher's LDA scheme, such distance is measured leveraging the covariance matrix. Although LDA is a linear analysis, we empirically find that it is a good criterion to identify ineffective channels.

Recall the notations in Section 2. $z_i^j$ is a column vector representing the flattened $j$-th channel of the output representation for image $x_i$ with $|z_i^j| = W' \times H'$. To evaluate Fisher's criterion for the $j$-th channel, let $N_k$ denote the number of $z_i^j$ with $y_{i,k} = 1$, and let $Z^j$ denote the matrix formed from the $N_k$ data points in class $k$. For the $k$-th class, we denote the average of the representations as

$\bar{\mathbf{z}}_k$, i.e., $\bar{\mathbf{z}}_k^j = \frac{1}{N_k} \sum_{i:\mathbf{y}_{i,k}=1} \mathbf{z}_i^j$. $\bar{\mathbf{z}}^j$ is the average over all $\{\mathbf{z}_i^j\}_{i=1}^N$. Finally, define $\bar{Z}_k^j$ as the matrix containing $N_k$ copies of $\bar{\mathbf{z}}_k^j$.

Given the notations above, for the $j$-th output channel, Fisher's linear discriminability can be computed as

$$J(Z_1^j, \ldots, Z_K^j) = \max_p \frac{p^\top S_b p}{p^\top S_w p}, \tag{6}$$

where $S_b = \sum_k (\bar{\mathbf{z}}_k^j - \bar{\mathbf{z}}^j)(\bar{\mathbf{z}}_k^j - \bar{\mathbf{z}}^j)^\top$ denotes the between-class variance and $S_w = \sum_k (Z_k^j - \bar{Z}_k^j)(Z_k^j - \bar{Z}_k^j)^\top$ denotes the within-class variance. As has been proved in (Fisher, 1936), the maximum value can be achieved when $p$ is the eigenvector corresponding to the largest eigenvalue of $S_w^{-1} S_b$ and thus, $J(Z_1^j, \ldots, Z_K^j)$ equals to the largest eigenvalue of $S_w^{-1} S_b$. By evaluating the Fisher's discriminability for the representations generated by each channel, we can determine the channels with the worst utility, which will be pruned to provide a better accuracy for the learning task.

### 3.5 EFFECTIVENESS OF SUPERVISED CHANNEL PRUNING

Now, we verify the effectiveness of the LDA-based supervised channel pruning algorithm. The experimental setup is the same as our characterization in Section 2.

We first verify the effectiveness of leveraging the Fisher's discriminability to identify ineffective channels. We use the first 6 VGG16 layers to form the FEN and try to prune the 32 output channels with the worst utility. We use 50 mini-batches of samples and the size of each mini-batch is 128, i.e., $N_{LDA} = 6.4 \times 10^3$, for the LDA-based supervised pruning. As in Figure 11(a), when 64 channels are pruned, with our method, 69.7% of the worst 32 channels can be pruned on average while only 50.3% can be pruned with the random pruning method. This translates to 33.5% reduction of the probability to get a bad channel when we randomly select a channel from the remaining ones. Similar results can be observed when we try to prune the 64 channels with the worst utility as in Figure 11(b).

Then, we explore the number of samples that are required for the LDA-based pruning. In Figure 12(a), to prune the 32 channels with the worst utility, we change the mini-batch number from 10 to 20 and 50. As we can see, very similar average values and standard deviations can be acquired on the pruning rate. A similar observation can be made if we want to prune the worst 64 channels. According to the complexity analysis in Appendix E, the computation complexity of the LDA-based pruning process scales in proportional to the number of samples. Therefore, the experimental results indicate the extra computation introduced by the pruning process is small.

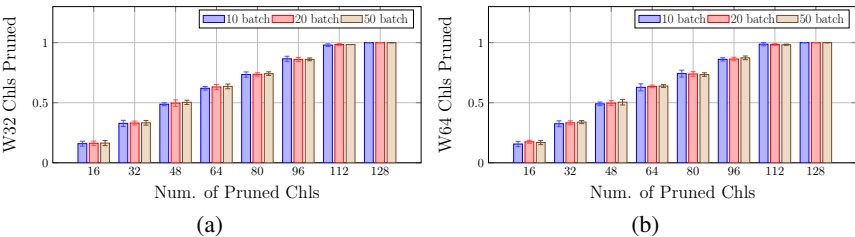

Figure 12: The number of samples required for the LDA-based pruning (mini-batch size is 128).

To demonstrate the effectiveness of supervised channel pruning, we set the layer of FEN to be 6 and the output depth to be 8. We compare the privacy and utility for the released representations in the following three settings:

- Random selection within the whole set of output channels.
- Channel pruning based on privacy and utility characterization results followed by random selection.
- Channel pruning based on privacy characterization and LDA followed by random selection.

In the pruning process, we prune the 64 channels with the worst utility identified by characterization or LDA and 32 channels with largest privacy loss identified by characterization. We run 20 experiments

with random selection for each setting and show the results in Figure 13(a). As we can see, while the average PSNR for random selection without pruning (the first setting) is 18.3 dB, the PSNRs of all the experiments for the second and third settings are less than the average value. Meanwhile, most of the points for the second and third settings are in the lower right corner in Figure 13(a), which indicates after pruning, we can achieve a better utility and less privacy leakage simultaneously. We also list the detailed statistics in Table 13(b). As we can see, compared with a random selection without pruning (the first setting), our LDA-based pruning strategy (the third setting) achieves on average 1.1% better accuracy and 1.25 dB smaller PSNR. Compared with the pruning strategy based on the characterization results (the second setting), our method achieves very similar accuracy (around 0.5%) with slightly less privacy loss (around 0.45 dB). Therefore, the effectiveness of our supervised pruning strategy is verified.

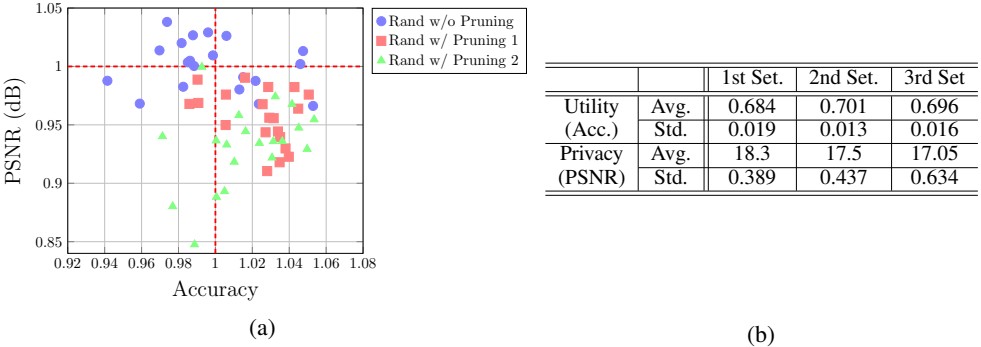

|  |  | 1st Set. | 2nd Set. | 3rd Set |
|---|---|---|---|---|
| Utility | Avg. | 0.684 | 0.701 | 0.696 |
| (Acc.) | Std. | 0.019 | 0.013 | 0.016 |
| Privacy | Avg. | 18.3 | 17.5 | 17.05 |
| (PSNR) | Std. | 0.389 | 0.437 | 0.634 |

(a)  (b)

Figure 13: Utility and privacy comparison for the three settings, including random selection without pruning (1st Set.), random selection after pure characterization-based pruning (2nd Set.) and random selection after LDA-based pruning.

## 4 DISCUSSION

In this section, we provide detailed discussions on the adversarial model adopted in the paper. According to the adversarial model we have defined in Section 3, the transformation induced by the FEN is assumed to be unknown to the attackers. This helps prevent more powerful attacks and enable a better privacy protection. However, because the FEN is derived from the pre-trained NNs, the structure and weights of which are also available to the attackers, we need to provide strategies to protect the anonymity of the FEN. In our framework, we consider the following two methods for the protection of the FEN:

- Build a pool of pre-trained NNs to enable FEN derivation from NNs. In our characterization framework, we use VGG16 as an example. The same procedure can be applied to VGG19 (Simonyan & Zisserman, 2014), ResNet (He et al., 2016), Inception (Szegedy et al., 2016). By enlarging the pool, it becomes harder for the attacker to guess how the FEN is derived.

- Apply the channel selection procedure to both output channels and intermediate channels. After the channel selection, the number of channels and the subset of selected channels in each layer become unknown to the attackers. Therefore, even if the attackers know the pre-trained NN, from which the FEN is derived, it becomes much harder to guess the channels that form the FEN.

One important requirement for the intermediate channel selection is that the utility is not sacrificed and the privacy loss is not increased. We verify the change of privacy and utility empirically. We take the first 6 layers of VGG16, including 4 convolution layers and 2 max-pooling layers, and set the depth of output channel to 8. We use CIFAR-10 dataset and the same ICN and IRN as in Section 2. We first gradually reduce the channel depth of the first convolution layer from 64 to 16. As shown in Figure 14, the privacy and utility are rarely impacted by the change of the channel depth of first convolution layer. Meanwhile, we can observe the dramatic reduction on the runtime.

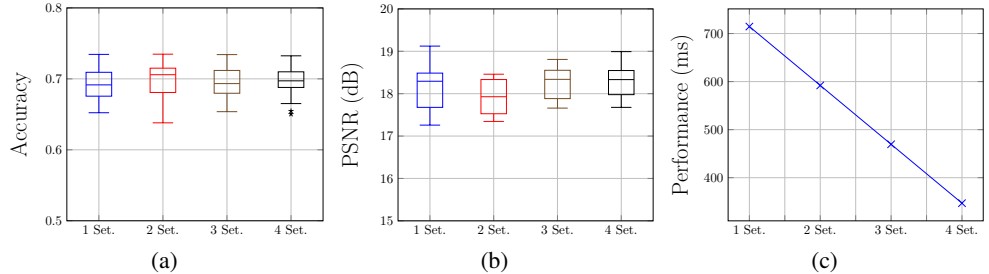

Figure 14: Impact of pruning the first layer of FEN on (a) accuracy, (b) privacy, and (c) local computation. The channel depths of the four convolution layers for the four settings are $\{64, 64, 128, 8\}$ (baseline), $\{48, 64, 128, 8\}$, $\{32, 64, 128, 8\}$, $\{16, 64, 128, 8\}$, respectively.

We then gradually reduce the channel depth for each convolution layer. As shown in Figure 15, after we reduce the channel depth for each layer to half of its original depth, we still get similar privacy and utility with a dramatic reduction of the runtime.

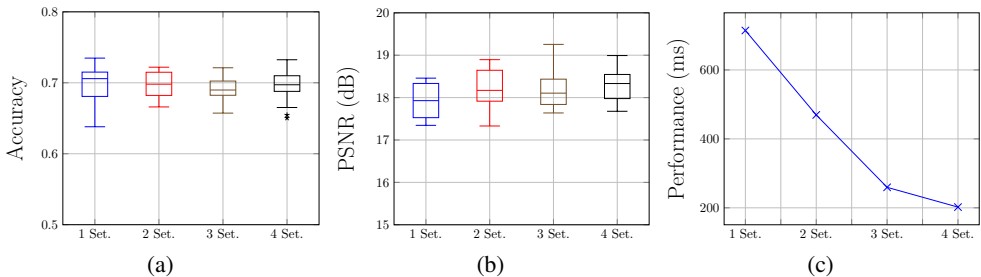

Figure 15: Impact of pruning each convolution layer step by step on (a) accuracy, (b) privacy, and (c) local computation. The channel depths of the four convolution layers for the four settings are $\{64, 64, 128, 8\}$ (baseline), $\{32, 64, 128, 8\}$, $\{32, 32, 128, 8\}$, $\{32, 32, 64, 8\}$, respectively

By channel selection for intermediate layers, even if the attackers can know the pre-trained NN that our FEN is derived from, it is still very hard to determine the number of layers for the FEN and the number of channels for each layer. In this way, the anonymity of the FEN can be well protected.

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

## A  USE CASES

PrivyNet is a flexible framework designed to enable cloud-based training while providing a fine-grained privacy protection. It can help to solve the problem of resource-constrained platforms, lack of necessary knowledge/experience and different policy constraints. One use case would be for modern hospitals, which usually hold detailed information for different patients. Useful models can be trained from the patients' data for disease diagnosis, prevention, and treatment. However, strong restrictions are usually enforced to the patients' data to protect their privacy. PrivyNet provides an easy framework for the hospitals to release the informative features instead of the original data. Meanwhile, it does not require too much knowledge and experience on DNN training. Another application would be for the pervasive mobile platforms, which have high capability to collect useful information for a person. The information can be used to help understand the person's health, habit, and so on. Because the mobile platforms are usually lightweight with limited storage and computation capability, PrivyNet can enable mobile platforms to upload the collected data to the cloud while protecting the private information of the person. Overall, PrivyNet is simple, platform-aware, and flexible to enable a fine-grained privacy/utility control, which makes it generally applicable for different end-users in different situations.

## B  CHARACTERIZATION SETTINGS

**Datasets** In our characterization, we use CIFAR-10 and CIFAR-100 datasets (Krizhevsky & Hinton, 2009). The CIFAR-10 dataset consists of 60000 $32 \times 32$ color images in 10 classes, with 6000 images per class. The images are separated into 50000 training images and 10000 test images. The CIFAR-100 dataset consists of images of objects belonging to 100 classes. For each class, there are 600 images with 500 training images and 100 test images. The size of each image is also $32 \times 32$.

**Networks and training settings** As an example, we derive the FEN from VGG16 (Simonyan & Zisserman, 2014), which is pre-trained on ImageNet dataset (Russakovsky et al., 2015), for the privacy and accuracy characterization. The full architecture of VGG16 is shown in Appendix C. We use CNN to construct $h$ for the image classification task and $g$ for the image reconstruction task. For $h$, we use the network from Tensorflow example[1], the architecture of which is shown in Appendix C. For $g$, we use the state-of-the-art generative NN architecture based on ResNet blocks (He et al., 2016), which has demonstrated good performance for different image recovery tasks, including super resolution (Ledig et al., 2016), denoising autoencoder (Dong et al., 2016), and so on. We follow (Ledig et al., 2016) and construct image IRN as shown in Figure 18. For each ResNet block cluster, there are 8 ResNet blocks, the structure of which is shown in Figure 18 as well. In our experiments, we follow the Tensorflow example and use gradient descent optimizer in the training process. For the image reconstruction task, the learning rate is set to 0.003 and the mini-batch size is set to be 128. We train in total 100 epochs. For the image classification task, the initial learning rate is set to 0.05 and mini-batch size is set to 128. The learning rate is dropped by a factor 0.1 at 100 and 200 epochs, and we train for a total of 250 epochs. For the data augmentation, we do normalization for each image and modify the brightness and contrast randomly following Tensorflow example[1].

## C  ARCHITECTURE OF VGG16 AND IGN

Before the characterization, we first determine the topology of the IRN to guarantee its capability to recover the original images for accurate privacy evaluation. The image recovery capability of IRN is mainly determined by the number of ResNet block clusters. We run the image reconstruction experiments on the representations generated by FENs with different topologies and keep a record of the quality change of the reconstructed images. As shown in Figure 19, the PSNR of the reconstructed images saturates with the increase of the number of ResNet block clusters. Therefore, in our experiments, we choose 2 ResNet block clusters, with 8 blocks as in each cluster. To understand the implications of the PSNR value, we show the reconstructed images with different PSNRs in Appendix F.

---

[1]http://www.tensorflow.org/tutorials/deep_cnn

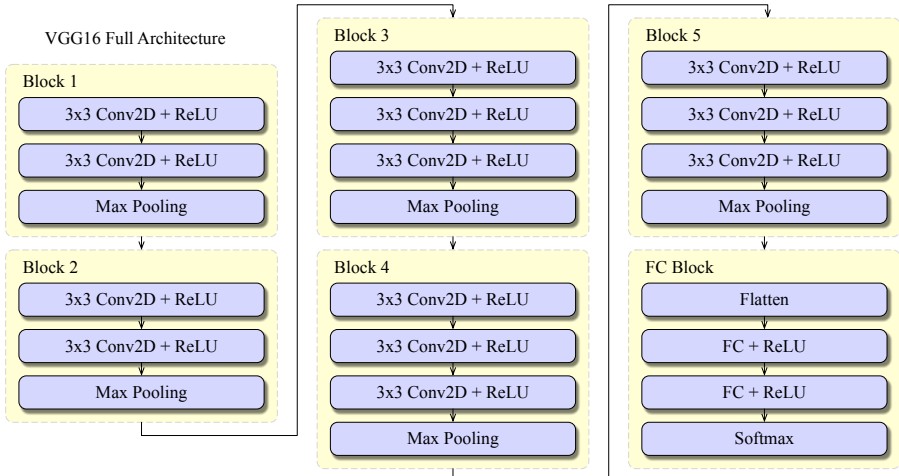

Figure 16: Full VGG16 architecture.

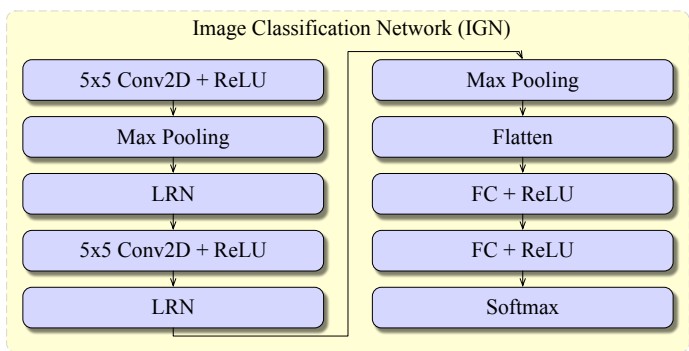

Figure 17: IGN architecture from Google example[1].

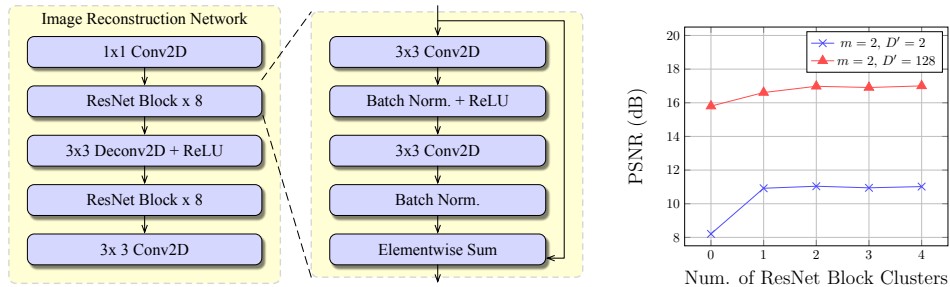

Figure 18: Architecture of the IRN and the ResNet block.  Figure 19: Determine IRN architecture.

# D  PERFORMANCE AND STORAGE PRE-CHARACTERIZATION FOR DIFFERENT PLATFORMS

Performance and storage characterization is realized by profiling different pre-trained NNs on the local platforms. In Figure 20(a) and 20(b), we show the performance profiling of VGG16 on a mobile class CPU and on a server class CPU. The change of storage requirement with the increase of VGG16 layers is also shown in Figure 20(c). As we can see, with the increase of the number of VGG16 layers, requirements on local computation and storage increase rapidly. Meanwhile, we observe that most of the computation comes from the convolution layers while for the storage, fully connected layers account for a significant portion. Especially with the increase of the size of the input image, fully connected layers will take an even larger portion of storage. For platforms with different computation

and storage configurations, the bottleneck may be different. Moreover, significant runtime difference can be observed for different platforms, which further indicates the necessity to have a framework that is flexible to consider the difference in local computation capability and storage.

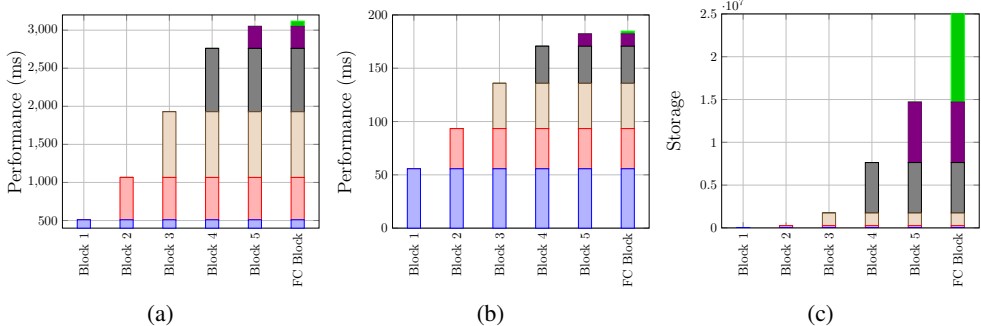

Figure 20: Performance profiling for VGG16 on (a) a mobile class CPU; (b) a server class CPU; and (c) storage profiling (batch size is 1 and input size is (32, 32, 3)).

## E   COMPLEXITY OF LDA-BASED CHANNEL SELECTION

The channel pruning process is finished locally before the channel selection. Compared with simply selecting channels from the whole set, the extra computation introduced in the process mainly consists of two parts. First, to enable LDA-based pruning, instead of evaluating $D'$ feature representations, more representations need to be computed. Secondly, the calculation of $S_b$, $S_w$, $S_w^{-1}$ and the eigenvalue of $S_w^{-1}S_b$ also incurs extra computation overhead.

Let $N_{LDA} = \sum_{k=1}^K N_k$ be the total number of samples required for the channel selection. The first part of the extra computation mainly comes from the last convolution layer in the FEN. Assume the dimension of each filter is $W_k \times H_k \times D_k$. Let $W_f \times H_f \times D_f$ be the dimension of the input to the last convolution layer. Then, we have $D_k = D_f$. For simplicity, we assume the dimension of each output channel of the last convolution layer is also $W_f \times H_f$. Also, assume that we need to select $D'$ channels out of the total $D_r$ channels. Then, the extra computation mainly comes from the convolution between the $D_r - D'$ filters, each of dimension $W_k \times H_k \times D_k$, with the input features of dimension $W_f \times H_f \times D_f$. Because for a given FEN topology, we can further have $W_f = \mathcal{O}(W')$ and $H_f = \mathcal{O}(H')$, consider $N_{LDA}$ samples, the extra computation is $\mathcal{O}(N_{LDA}W'H'W_kH_kD_f(D_r - D'))$.

The second part of the extra computation is mainly determined by the number of samples $N_{LDA}$ and the dimension of the output representations $W' \times H'$. To get $S_b$, the complexity is $\mathcal{O}(KW'^2H'^2)$, while to get $S_w$, the complexity is $\mathcal{O}(N_{LDA}W'^2H'^2)$. To compute $W^{-1}$ and the largest eigenvalue of $W^{-1}B$, the size of which are both $W' \times H'$, the complexity is $\mathcal{O}(W'^3H'^3)$. Therefore, the complexity of the second part of the computation is thus $\mathcal{O}((K + N_{LDA})W'^2H'^2 + W'^3H'^3)$.

The total computation complexity thus becomes $\mathcal{O}(N_{LDA}W'H'W_kH_kD_f(D_r - D') + (K + N_{LDA})W'^2H'^2 + W'^3H'^3)$. While $D_f, W_k, H_k, D_f, D', K$ are determined by the characteristics of the FEN and the target learning task, $N_{LDA}$ becomes the key factor that determines the extra computation induced by the learning process. As we will show later, because usually small $N_{LDA}$ is sufficient to achieve good pruning results, the overall introduced computation overhead is small.

## F   EXAMPLE OF RECONSTRUCTED IMAGES

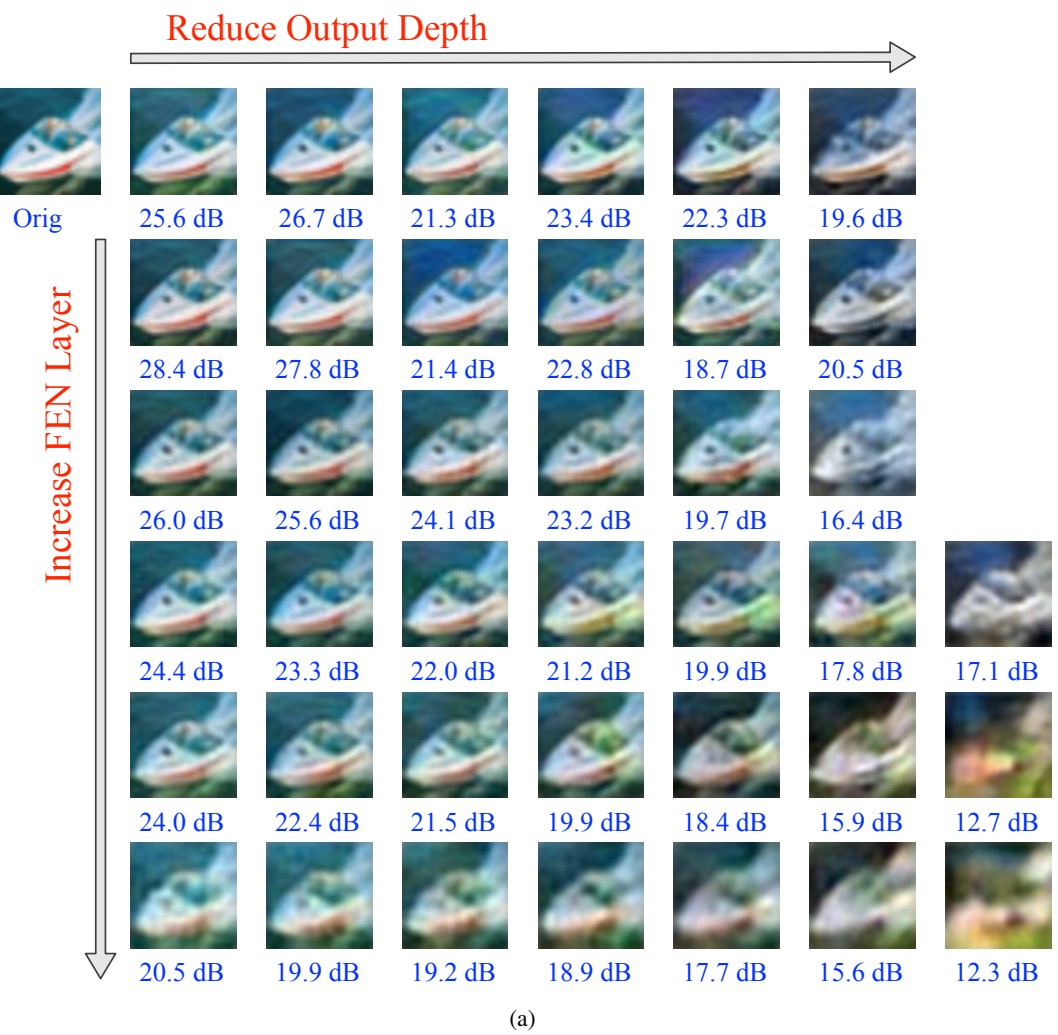

(a)

Figure 21: Example 1: Impact of number of layers and output depth on the quality of reconstructed images: the output depths are $\{64, 32, 16, 8, 4, 2\}$ for the first three rows and are $\{128, 64, 32, 16, 8, 4, 2\}$ for the last three rows (original figures are selected from CIFAR-10 dataset).

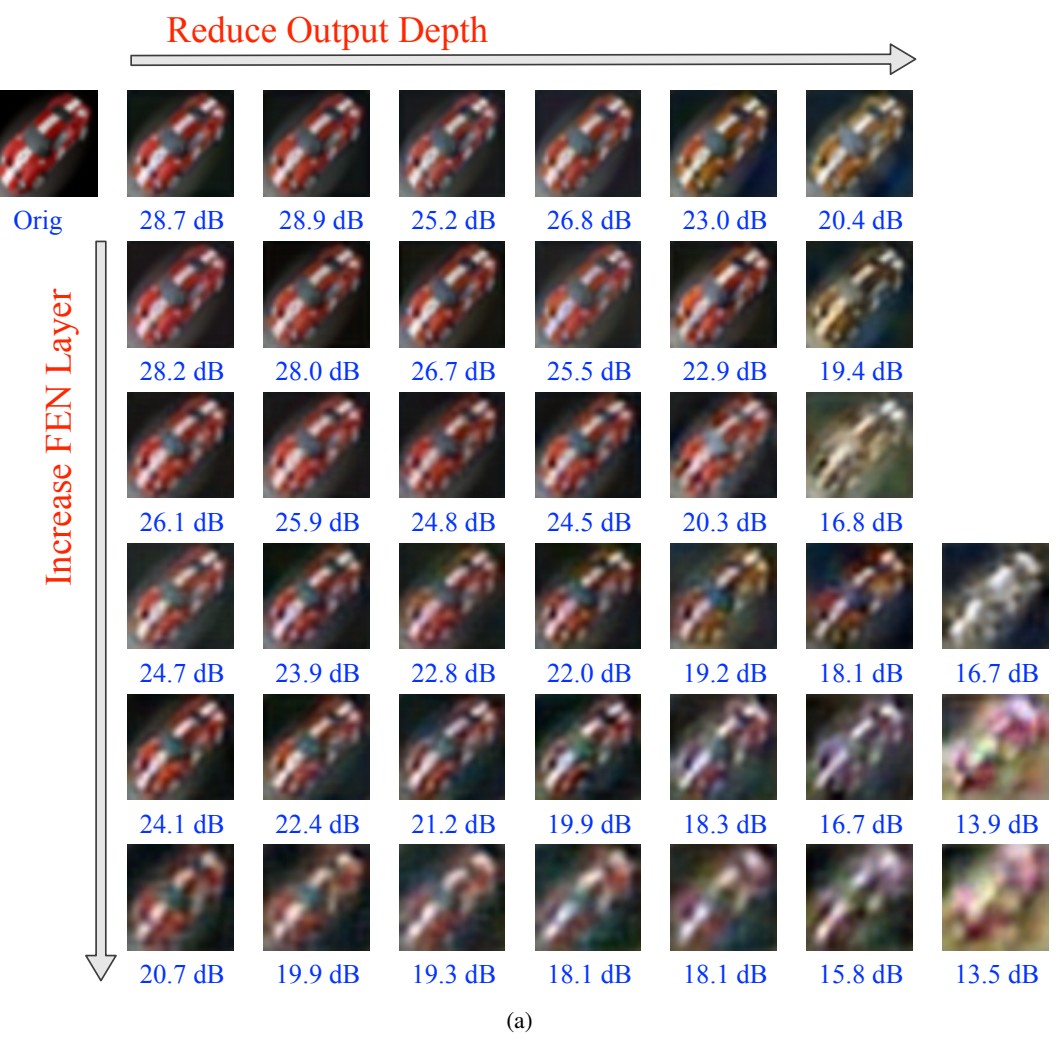

(a)

Figure 22: Example 2: Impact of number of layers and output depth on the quality of reconstructed images: the output depths are $\{64, 32, 16, 8, 4, 2\}$ for the first three rows and are $\{128, 64, 32, 16, 8, 4, 2\}$ for the last three rows (original figures are selected from CIFAR-10 dataset).

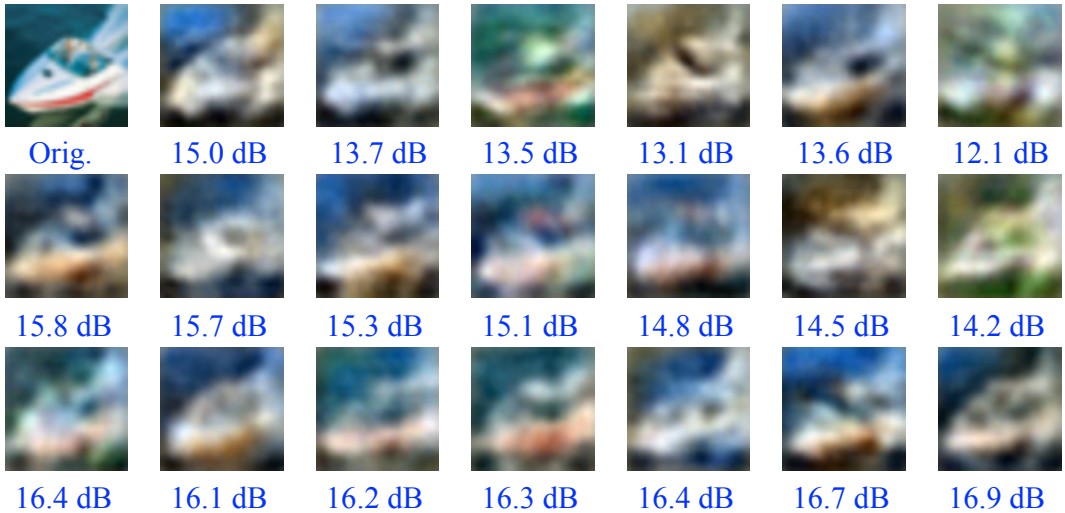

Figure 23: Example 3: Impact of output channel selection on the quality of reconstructed images, $m = 6$, $D' = 4$.

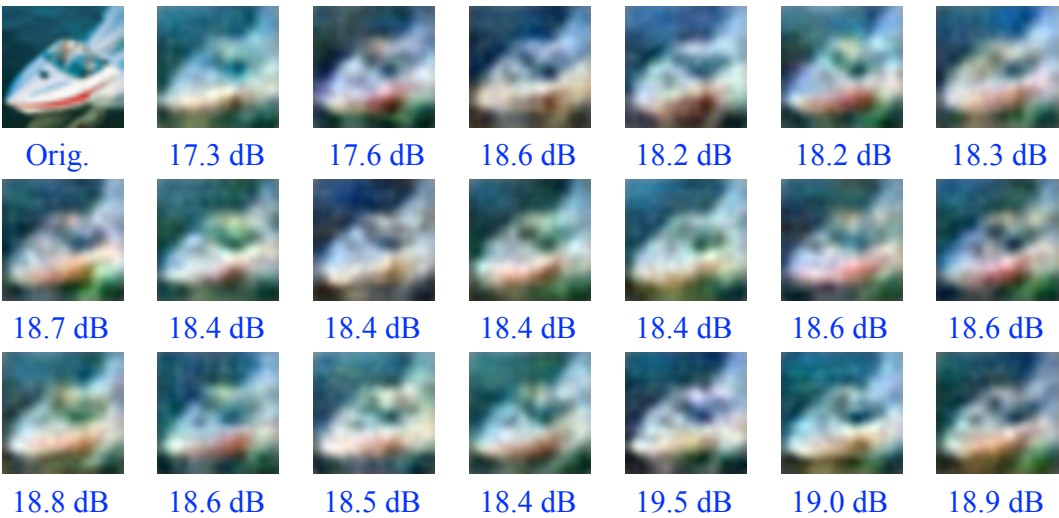

Figure 24: Example 4: Impact of output channel selection on the quality of reconstructed images, $m = 6$, $D' = 16$.

