# OpenReview forum: "PrivyNet: A Flexible Framework for Privacy-Preserving Deep Neural Network Training"
_ICLR.cc/2018/Conference — Reject_

### Official Review · AnonReviewer3 · 2017-11-26
**This is a generally interesting paper introducing some useful formulations of utility and privacy loss**

**Rating:** 6
**Confidence:** 5

**Review:**

1. This is an interesting paper - introduces useful concepts such as the formulation of the utility and privacy loss functions with respect to the learning paradigm
2. From the initial part of the paper, it seems that the proposed PrivyNet is supposed to be a meta-learning framework to split a DNN in order to improve privacy while maintaining a certain accuracy level
3. However, the main issue is that the meta-learning mechanism is a bit ad-hoc and empirical - therefore not sure how seamless and user-friendly it will be in general, it seems it needs empirical studies for every new application - this basically involves generation of a pareto front and then choose pareto-optimal points based on the user's requirements, but it is unclear how a privy net construction based on some data set considered from the internet has the ability to transfer and help in maintaining privacy in another type of data set, e.g., social media pictures

---

### Official Review · AnonReviewer1 · 2017-11-27
**I am concerned about the notion of privacy used in the paper.**

**Rating:** 5
**Confidence:** 3

**Review:**

Summary: The paper studies the problem of effectively training Deep NN under the constraint of privacy. The paper first argues that achieving privacy guarantees like differential privacy is hard, and then provides frameworks and algorithms that quantify the privacy loss via Signal-to-noise ratio.  In my opinion, one of the main features of this work is to split the NN computation to local computation and cloud computation, which ensures that unnecessary amount of data is never released to the cloud.

Comments: I have my concerns about the effectiveness of the notion of privacy introduced in this paper. The definition of privacy loss in Equation 5 is an average notion, where the averaging is performed over all the sensitive training data samples. This notion does not seem to protect the privacy of every individual training example, in contrast to notions like differential privacy. Average case notions of privacy are usually not appreciated in the privacy community because of their vulnerability to a suite of attacks.

The paper may have a valid point that differential privacy is hard to work with, in the case of Deep NN. However, the paper needs to make a much stronger argument to defend this claim.

---

### Official Review · AnonReviewer2 · 2017-11-27
**Doubts about the Privacy of the Method**

**Rating:** 3
**Confidence:** 3

**Review:**

1. Paper summary

This paper describes a technique using 3 neural networks to privatize data and make predictions: a feature extraction network, an image classification network, and an image reconstruction network. The idea is to learn a feature extraction network so that the image classification network performs well and the image reconstruction network performs poorly.


2. High level paper - subjective

I think the presentation of the paper is somewhat scattered: In section 2 the authors introduce their network and their metric for utility and privacy and then immediately do a sensitivity analysis. Section 3 continues with a sensitivity analysis now considering performance and storage of the method. Then 2.5 pages are spent on channel pruning.
I would have liked if the authors spent more time justifying why we should trust their method as a privacy preserving technique (described in detail below).
The authors clearly performed an impressive amount of sensitivity experiments. Assuming the privacy claims are reasonable (which I have some doubts about below) then this paper is clearly useful to any company wanting to do privacy preserving classification. At the same time I think the paper does not have a significant amount of machine learning novelty in it.


3. High level technical

I have a few doubts about this method as a privacy-preserving technique:
- Nearly every privacy-preserving technique gives a guarantee, e.g., differential privacy guarantees a statistical notion of privacy and cryptographic methods guarantee a computational notion of privacy. In this work the authors provide a way to measure privacy but there is no guarantee that if someone uses this method their data will be private, by some definition, even under certain assumptions.
- Another nice thing about differential privacy and cryptography is that they are impervious to different algorithms because it is statistically hard or computationally hard to reveal sensitive information. Here there could be a better image reconstruction network that does a better job of reconstructing images than the ones used in the paper.
- It's not clear to my why PSNR is a useful way to measure privacy loss. I understand that it is a metric to compare two images that is based on the mean-squared error so a very private image should have a low PSNR while a not private image should have a high PSNR, but I have no intuition about how small the PSNR should be to afford a useful amount of privacy. For instances, in nearly all of the images of Figures 21 and 22 I think it would be quite easy to guess the original images.


4. 1/2 sentence summary

While the authors did an extensive job evaluating different settings of their technique I have serious doubts about it as a privacy-preserving method.

---

### Decision · Program_Chairs · 2018-01-29
**ICLR 2018 Conference Acceptance Decision**

**Decision:**

Reject

**Comment:**

Reviews are marginal.
I concur with the two less-favorable reviews that the metrics  for privacy protection are not sufficiently strong for preserving privacy.